# Cross-Language Perception of Lexical Tones by Nordic Learners of Mandarin Chinese

**Man Gao**

Department of Chinese, School of Language, Literatures and Learning, Dalarna University, 791 88 Falun, Sweden; mao@du.se

**Abstract:** While existing cross-language studies on the perception of non-native tones primarily focus on naïve listeners, this study addresses an obvious gap by investigating learners from diverse language backgrounds. Specifically, it investigates Mandarin tone perception in two groups of learners from Nordic languages, Swedish (a pitch-accent language), and Danish (a non-tonal language), as well as in a group of native Chinese speakers. Analysis of their performance in tone identification task revealed a slight advantage for Swedish learners, implying the influence of their pitch accent background in learning Mandarin tones. However, both Swedish and Danish learners who excelled in the tone identification task exhibited similar perception of within-category tonal variations but differed from native Chinese speakers. Additionally, the study found that the presence of length contrast, a prosodic feature in the learners' native languages, significantly influences their perception of Mandarin tones.

**Keywords:** tone perception; Mandarin Chinese; Nordic languages; L2 learners; pitch-accent language

## 1. Introduction

There exists very extensive literature on the topic of cross-language speech perception, which investigates how sounds in a language are recognized and parsed by non-native speakers. Research of this type is dominated by studies of the perception of speech segments (i.e., vowels and consonants), which have shown that listeners' native language (or L1) affects their perception of non-native sounds and sound contrasts. Over the past two decades, we have seen an increase in research on the perception of suprasegmental features (e.g., tones, pitch accents, and stress). The L2 perception of Mandarin tones has attracted most attention because the tonal system in Mandarin is a major obstacle for many categories of adult learners of Mandarin Chinese (Sun 1998; Ke and Shen 2003).

The majority of the work on the perception of Mandarin tones has been carried out with listeners whose L1 is English (e.g., Kiriloff 1969; Broselow et al. 1987; Wang et al. 1999; Hao 2012). There are a number of studies, however, that have examined the perception of Mandarin tones by listeners from other non-tonal language backgrounds, such as French and Dutch (e.g., Hallé et al. 2004; So and Best 2014; Liu et al. 2017), as well as those from another tone language, such as Cantonese and Hmong (e.g., So and Best 2010; Hao 2012; Wang 2013), Thai and Vietnamese (Tsukada 2019), and Burmese, Thai, and Vietnamese (Tsukada 2022). Studies have focused on the perceptions of naïve listeners and of Mandarin learners in roughly equal proportions. Some of these studies have used natural speech produced by native Mandarin speakers in the listening tasks (e.g., Broselow et al. 1987; So and Best 2010; Hao 2012), while others have utilized synthesized or manipulated speech (e.g., Blicher et al. 1990; Hallé et al. 2004; Peng et al. 2010). What has been established by these studies can be summarized as follows: firstly, Mandarin tone perception is challenging; secondly, while native speakers of Mandarin perceive tones categorically, non-native speakers perceive these tones either continuously or psychophysically (e.g., Hallé et al. 2004; Peng et al. 2010; Liu et al. 2017); and thirdly, although there are a number of contributory factors, the role

of a listener's native language plays a very important role in the perception of Mandarin tones (e.g., So and Best 2010; Peng et al. 2010; Hao 2012).

*1.1. Perception of Mandarin Tones by Non-Native Listeners from Diverse Language Backgrounds*

Earlier work has demonstrated that it is a formidable task for non-natives to perceive the four Mandarin tones correctly, both for naïve listeners and second-language (hereafter L2) Mandarin learners (e.g., Kiriloff 1969; Broselow et al. 1987; Shen 1989; Wang et al. 1999; So and Best 2010; Hao 2012; Tsukada 2019). Among the four lexical tones of Mandarin, the level of perceptual difficulty varies. For English-speaking learners, Tone 2 (with all tones hereafter referred to as Tx) and Tone 3 have often been reported as particularly difficult, whereas Tone 4 has been found to be the least challenging (Kiriloff 1969; Broselow et al. 1987; Chen 1997; Sun 1998; Hao 2012). Analysis of English listeners' error patterns suggests that the confusion lies primarily in two pairs of tones: T2–T3 and T1–T4 (Chen 1997; Sun 1998; So and Best 2010; Hao 2012). Nonetheless, challenges in perceiving Mandarin tones can be surmounted through training and the accumulation of learning experiences, as indicated by findings from previous studies (e.g., Hao 2012, 2018; Shen and Froud 2016; Tsukada and Han 2019). In their 2019 study, Tsukada and Han compared how native Korean speakers, with varying proficiency levels in Mandarin Chinese, perceived six Mandarin tone pairs by native Korean speakers. The findings revealed that advanced Mandarin learners generally outperformed naïve listeners, exhibiting native-like perception for most tone pairs under examination, with the exception of T1–T2.

Thus far, few studies (So and Best 2010; Wang 2013; Tsukada 2019) have sought to compare the perception of Mandarin tones by listeners from different L1 language types. The perception of Mandarin tones was investigated among naïve listeners (So and Best 2010) and novice L2 Chinese learners (Wang 2013) with backgrounds in pitch-accent language (Japanese), non-tonal language (English), and another tone languages (Hong Kong Cantonese and Hmong). A more recent study by Tsukada (2019) also compared Mandarin tone perception among three groups of naïve listeners who were native speakers of Australian English (non-tonal language), Thai (tone language), and Vietnamese (tone language). The findings across these three studies revealed variations in the overall perceptual abilities of listeners from diverse language backgrounds. In the study by So and Best (2010), listeners from tone (Cantonese) and pitch accent (Japanese) languages exhibited better performance compared with English (a non-tonal language) listeners. However, Wang (2013) reported a different outcome, with English and Japanese listeners outperforming those from a tone language background (Hmong). Meanwhile, Tsukada (2019) identified distinctive perception patterns in discrimination tasks involving Chinese tone pairs, with Thai and Vietnamese (tone language) listeners demonstrating differences not only from Australian listeners, but also from each other. Both Wang (2013) and Tsukada (2019) underscore a common perspective, suggesting that the pitch accents or tonal system in listeners' native languages may not necessarily offer any advantages in perceiving non-native tones, when compared with listeners from non-tonal language backgrounds. While these studies provide valuable insights into cross-linguistic perception of tones by naïve listeners, there is a paucity of studies addressing this issue within the context of L2 learners from diverse language backgrounds, highlighting a gap that requires further investigation.

*1.2. Effect of Listeners' Native Languages on Tone Perception*

It is well supported by previous experimental studies that listeners' native language affects their perception of an L2 language. In fact, a number of factors have been identified as influencing the perception of non-native speech, such as listeners' age, experience with the target language, and their native language (L1) (Best 1995; Flege 1995; Yamada 1995). There is a large body of work that concerns not only tone perception, but also the perception of consonants and vowels, which provides clear, strong, and well-documented evidence to suggest that the perception of non-native speech is significantly influenced by L1 (Wenk 1986; Odlin 1989; Jenkins and Yeni-Komshian 1995). Two influential theoretical models

which were originally proposed to account for the perception of non-native speech at a segment level have also been extended to the perception of prosodic categories. The Speech Learning Model (SLM) (Flege 1995, 2007) hypothesizes that for L2 learners, if a novel speech segment in L2 is perceptually similar to a neighboring sound in the L1, assimilation of the L2 sound to L1 will occur. The SLM has also been successfully applied to test the perception of Mandarin tones by L2 learners in Hao (2014). The other important model, the Perceptual Assimilation Model (PAM) (Best 1995; So and Best 2008), suggests that naïve non-native listeners assimilate segments and suprasegments of another language to those that are most similar to the ones in their native phonetic system. In the case of Mandarin tones, naïve English listeners assimilate them to the intonational categories in English (So and Best 2008). More recent studies (Chen et al. 2020) investigating the perception of Thai tones have contributed additional evidence supporting the PAM. These studies highlight the significance of listeners' L1 phonological and phonetic influences on the perception of non-native tones. To sum up, both the PAM and SLM acknowledge the importance of native language for the perception of non-native speech; the phonetic categories and phonological structure in the native language exert a strong influence on how the listener perceives non-native speech.

### 1.3. Categorical Perception of Mandarin Tones

"Categorical perception" refers to a phenomenon concerning the non-continuous perception of continuous series of stimuli. When presented with the continuum, listeners identify the stimulus as one category at one end of the continuum, and at some point in the continuum, swiftly change to the other category at the other end, rather than perceiving the physically continuous stimuli as a gradual continuum (Repp 1984). One consequence of this is that variation between two separate categories is easier to detect than variation within one category. About half a century ago, scholars began investigating the categorical perception of Mandarin tones (Wang 1976). Wang's (1976) experimental study, which employed stimuli from a T1–T2 continuum, offered evidence that while native speakers perceive Mandarin tones categorically, American listeners perceive them non-categorically. This finding is further supported by a subsequent study (Leather 1987) in which both Dutch and English listeners were less categorical than native speakers when perceiving Mandarin tones. More recently, Hallé et al. (2004) conducted a cross-language study on the categorical perception of lexical tones in Taiwanese Mandarin. An eight-step continuum was created for each of the T1–T2, T2–T4, and T3–T4 contrasts. The results of the identification and discrimination tasks completed by Taiwan Mandarin and naïve French listeners revealed the expected pattern. Native Taiwanese listeners perceived the tones nearly categorically, whereas the French listeners did so psychophysically. Peng et al. (2010) also conducted a comparative categorical perception study of Mandarin tones, which included native Chinese, German (non-tonal language), and Cantonese (tone language) speakers. Further evidence was found, in line with previous studies, that both native Mandarin and Cantonese listeners perceive Mandarin tones more categorically than German listeners, who presented a psychophysical perception. What the study also noted was that the Cantonese listeners' perception of Mandarin tones was influenced by the tonal system of their native language (Peng et al. 2010). A more recent study by Liu et al. (2017) replicated elements of these previous studies and came to a similar conclusion that Dutch naïve listeners' perception of Mandarin tones is psychophysical, but native Mandarin listeners' is categorical. However, almost all studies of this type have included only naïve listeners, leaving unanswered the question of whether learners of Mandarin Chinese might be able to perceive tonal variations categorically provided they have some experience with and exposure to Mandarin.

### 1.4. Current Study

Although the perception of Mandarin tones has garnered much scholarly attention, there is still very little published work that focuses on the tone perception of Mandarin by listeners from a pitch-accent language background other than Japanese (So and Best

2010; Wang 2013), and also in the context of Mandarin L2 learners. This study aims to contribute to the existing knowledge on Mandarin tone perception by L2 learners from two different and less studied language backgrounds: Swedish, a pitch-accent language, and Danish, a non-tonal language. Going beyond a mere emphasis on tone perception, we are also intrigued to investigate the sensitivity of highly proficient L2 Mandarin learners to the more subtle within-category tonal changes, as previous studies suggested that they could exhibit a native-like perception pattern (Shen and Froud 2016; Tsukada and Han 2019).

The research questions formulated for the current study are the following: (1) how do L2 Mandarin learners from a pitch-accent language background differ from those with a non-tonal language background in terms of accuracy rate and error patterns when perceiving the four Mandarin tones?; (2) do L2 learners, who can identify Mandarin tones with exceptional accuracy, demonstrate a comparable ability to perceive subtle tonal differences, i.e., within-category variations, to native Mandarin speakers? Furthermore, this study seeks to investigate the influence of learners' L1 prosodic features on the perception of Mandarin tones, in particular duration. Previous studies (Blicher et al. 1990; Chen et al. 2017) reported the duration effect on perception of subtle tonal variations by native Chinese speakers and English-speaking listeners. However, studies involving participants from languages characterized by length contrast are absent. It is noteworthy that Swedish and Danish, despite differing in pitch systems, share similarities in length contrast, a feature we will elaborate on in the next section. Thus, the third research question is: (3) to what extend does the L1 prosodic feature, specifically length contrast, impact the perception of within-category tonal variations in Mandarin Chinese?

*1.5. Mandarin Chinese vs. Swedish vs. Danish*

This section outlines some of the main prosodic features of Mandarin Chinese, Swedish and Danish, which are relevant to the current investigation. In this article, the term 'prosodic features' is used in a broad sense that includes tone, intonation, accent, stress, and length, as suggested by Fox (2000).

Mandarin Chinese is classified as a tone language in which the pitch contours are used to distinguish lexical meanings. Every monosyllable carries one of Tones 1–5, the five lexical tones as they are conventionally known. T1 is the only level tone that stays at high pitch level. T2 to T4 are contour tones, which are also referred to as rising, low, and falling tones, respectively, in accordance with the shape of the pitch contours. Tone 5 is commonly known as the neutral tone, which is only found in unstressed weak syllables. According to Chao (1968), the syllable preceding the weak syllable determines its pitch contour. Tones in Mandarin Chinese also vary in other aspects, such as duration (Chao 1968; Howie 1976; Tseng 1990, etc.) and amplitude (Ho 1976; Fu and Zeng 2000). Apart from Tone 5, the duration of the other four tones in citation form displays this pattern: T3 is the longest, followed by T2, T1, and T4. When produced in context, the tones are much shortened and the duration differences among tones become marginal (Howie 1976). Another important prosodic feature relevant to this study is the length contrast of segments. Among the six vowels (including the retroflex vowel [ɚ]) and 22 consonants (including three glides) in Mandarin Chinese (Duanmu 2007), there exists no short–long contrast like [k] and [kː] in Swedish or [i] and [iː] in Danish.

Swedish is classified as a pitch-accent language. The two pitch accents in Swedish, namely, Accent 1 (or 'acute') and Accent 2 (or 'grave'), can only be found in accented syllables. The schematic graph in Figure 1 illustrates the pitch contours of both accents in Central Swedish: Accent 1 is represented as a single falling tone with one peak, denoted as HL, whereas Accent 2 is characterized by two peaks that signal the primary and secondary stress of a word, represented as HLHL (Malmberg 1963; Bailey 1988). Just like lexical tones in Mandarin Chinese, these two accents can be used to contrast meaning of words that otherwise appear identical at the segmental level. Nevertheless, Accent 2 is only seen in words that have more than one syllable (Elert 1981). Swedish has approximately 350 to

500 minimal word pairs that contrast pitch accents, a tiny fraction of the language's total vocabulary (Elert 1971).

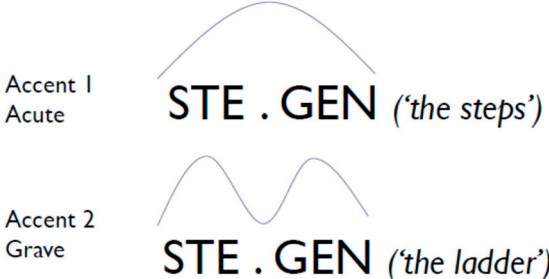

**Figure 1.** Representation of two pitch accents in Central Standard Swedish (Adopted from Engstrand (1997, p. 62)).

Swedish is well known for its quantity distinctions in its sound inventory, i.e., how vowels and consonants can differ in duration. Among its 17 vowels, there are 8 long vowels that are paired with 9 short vowels (Engstrand 1990; Riad 2014), although the long–short pairs also differ in vowel quality to some extent, e.g., [oː] vs. [ɔ] and [ɑː] vs. [a]. Some consonants can be either short or long, such as in words like *mat* [mɑːt] 'food' vs. *matt* [matː] 'feeble', depending on the preceding vowel and syllable type. In an oversimplified account, the vowel and consonant lengths are in complementary distribution in stressed syllables: a long vowel can only be followed by short consonant, or a long consonant preceded only by a short vowel. A detailed phonological account on Swedish quantity was published by Riad (2014). There have been multiple previous studies that investigated the perceptual cues to the length contrast in Swedish, among which Hadding-Koch and Abramson (1964) reported that native Swedish speakers relied mainly on durational contrast to distinguish vowel pairs.

As another Nordic language, Danish is similar to Swedish and Norwegian in many aspects. Nevertheless, Danish distinguishes itself from the other two languages with a unique prosodic feature—stød. Stød is described in the literature as 'a kind of creaky voice' or 'a laryngealization', and can contrast meaning between words in minimal pairs, e.g., *ham* [hɑmˀ] 'slough' (with stød [ˀ]) vs. *ham* [hɑm] 'him' (without stød) (Grønnum 1998a, p. 103; Grønnum and Basbøll 2001, p. 230). Stød is found in stressed syllables that contain either a long vowel or a short vowel followed by a sonorant consonant (Grønnum and Basbøll 2001). Although it is generally agreed among scholars that the Danish stød is historically related to the pitch accents in Swedish and Norwegian (Gårding 1977; Riad 1998), there is a long tradition of treating them separately (Basbøll and Kiefer 1972; Grønnum and Basbøll 2001; Riad 2006): pitch accent is understood as a tonal pattern but stød is not. Nevertheless, Danish has intonation at the sentence level and above, though the intonation pattern of Danish is considered rather "flat" and "less expressive" when compared with Swedish (Grønnum 1998b, p. 151). The topic of stød and intonation in Danish has been examined more extensively elsewhere (Basbøll 1985, 1988; Grønnum 1998b). With respect to vowel length contrast, Danish resembles its neighbors. Out of 16 monophthongal vowels in the Danish sound system, 13 vowels can be either short or long in words (e.g., *kugle* [kʰuːlə] 'ball' vs. *kulde* [kʰulə] 'cold') (Grønnum 1998a). Consonants in Danish, on the other hand, do not contrast in length.

## 2. Experiment 1

### 2.1. Methodology

#### 2.1.1. Participants

In total, 27 native Swedish speakers (8F, 19M) and 32 native Danish speakers (21F, 11M) who enrolled in beginner-level Chinese courses were recruited from universities in Sweden and Denmark to participate in the listening task. Sixteen native Chinese speakers studying

as international or exchange students (13F, 3M) at Swedish universities were also invited to participate as a control group. The Swedish and Danish participants were native speakers of Swedish and Danish, and none of them had any knowledge of another tone language except Mandarin Chinese. None of the participants reported being speakers of both Swedish and Danish. However, since both languages belong to the North Germanic language family, they share a considerable degree of mutual intelligibility. These two groups of participants were studying Mandarin Chinese at two different universities in Sweden and Denmark. All of them had spent at least 20 h per week learning Mandarin for a minimum of 4 months at the time when they participated in the listening tasks. Their Chinese proficiency level was A1 or A2 in the CEFR framework (Common European Framework of Reference for Languages: Learning, Teaching, Assessment), according to their teachers' assessments. In other words, they were considered basic users of Chinese, demonstrating fundamental spoken skills that enable them to engage in simple, everyday conversations.

### 2.1.2. Material and Procedure

All participants participated in a simple four-alternative forced choice identification task, which included 45 monosyllabic words produced by a native Mandarin speaker. The listening stimuli included 40 monosyllables, including ten syllables (ba, pao, fa, ge, mo, pi, tan, wan, ya, and yi) with four lexical tones, and 5 fillers (3 at the beginning and 2 at the end of the task). The choice of these syllables was based on two criteria: (a) the syllables were composed of consonants and vowels that exist in Danish, Swedish, or English, so that the participants could focus their attention on identifying tones in the listening task; and (b) the syllables carried all four Mandarin tones that correspond to existing morphemes, i.e., no accidental gaps.

The speech stimuli were produced by a female speaker whose native language is Beijing Mandarin. The recording took place in an anechoic chamber, where the speaker read the monosyllabic tokens in randomized order. The speech was captured with a Brüel and Kjær microphone which was placed at the distance of approximately 30 cm in front of the speaker's mouth, with a sampling rate of 44.1 kHz, bit depth of 16 Hz, and subsequently recorded to a hard drive. Praat (Boersma and Weenink 2018) was used to process the recording and prepare the sound file for use in the screening experiment.

Except for the five filler tokens, the 40 target syllables were randomized again before being presented to the three language groups. Every participant completed the listening task in a quiet environment. They were provided with detailed verbal and written instructions on how to complete the task: after listening to a Mandarin syllable, they were requested to identify the tone as Tone 1, Tone 2, Tone 3, or Tone 4. They could only listen to a token once and had six seconds to respond before the next token; however, in a similar listening task reported in So and Best (2010), the participants took less than four seconds to respond.

### 2.2. Results

The accuracy score of the identification task was computed for every participant, and the mean accuracy score was computed for every language group. Native Mandarin speakers all achieved a 100% correct rate, while Swedish learners performed better than the Danish learners (89.9% vs. 84.8%).

The Kruskal–Wallis one-way ANOVA was employed to compare the overall accuracy scores among the native Chinese group and the two learner groups, revealing a statistically significant difference ($\chi^2(3) = 28.00$, $p < 0.01$). Subsequent post hoc pairwise tests indicated that both learner groups exhibited significantly lower accuracy than the native Chinese group. However, the difference between the performance of Swedish and Danish learners was not statistically significant ($p = 0.08$), falling just outside the conventional threshold ($p < 0.05$). Comparison of scores across the four lexical tones revealed that the Swedish learners outperformed the Danes on every tone, particularly excelling on T2. Both groups displayed similar patterns: T1 and T2 were more challenging than the other two tones,

especially T2 (84.4% for Swedes and 73.1% for Danes). Within each learner group, the varied scores for the four tones were also subjected to the Kruskal–Wallis one-way ANOVA. For the Swedish group, no statistically significant difference ($\chi^2(3) = 4.69$, $p = 0.20$) was found. However, a statistically significant difference was found for the Danish group ($\chi^2(3) = 17.14$, $p < 0.01$), and subsequent post hoc pairwise tests indicated that the T2 score was significantly lower than T3 and T4.

Analysis of the perceptual errors revealed the patterns of tonal confusion for the two learner groups, as illustrated in Table 1. The most common tonal confusion for the Swedish group was from T1–T2 pair, which means that T1 is most likely to be misperceived as T2, and vice versa (about 10% for each tone). It was also observed that T3 was more likely to be misperceived as T2 (4.4%), but T4 was more likely to be misperceived as either T1 (2.6%) or T2 (2.6%). For the Danish group, tonal confusion was mostly found for the T2–T3 pair: T2 was most likely to be misperceived as T3 (15.3%) and vice versa, with a proportion of 7.5%. T1 was most likely to be misperceived as T2 (12.5%), and T4 was most likely to be misperceived as T2 as well (4.1%).

**Table 1.** Error matrix of the four Mandarin Tones by Swedish (blue) and Danish (red) listeners. Percentages in bold represent the accuracy of correctly identified tones.

| Target \ Response | | Tone 1 | Tone 2 | Tone 3 | Tone 4 | No Response |
|---|---|---|---|---|---|---|
| Tone 1 | SWE | **88.1%** | 10% | | 1.9% | |
| | DAN | **83.4%** | 12.5% | 1.3% | 2.8% | |
| Tone 2 | SWE | 9.6% | **84.4%** | 4.8% | 1.1% | |
| | DAN | 9.1% | **73.1%** | 15.3% | 2.2% | 0.3% |
| Tone 3 | SWE | 1.5% | 4.4% | **93.3%** | 0.7% | |
| | DAN | 2.8% | 7.5% | **89.4%** | 0.3% | |
| Tone 4 | SWE | 2.6% | 2.6% | 1.1% | **93.7%** | |
| | DAN | 1.3% | 4.1% | 1.6% | **93.1%** | |

Lastly, individual listeners' scores were checked for both groups. These ranged between 60% and 100%. Sixteen out of twenty-seven Swedish listeners scored 90% or more, and half of them achieved 100% accuracy. Only thirteen out of thirty-two Danish listeners scored 90% or more, and only four achieved a full accuracy score.

*2.3. Discussion*

The first experiment addressed the first research question, examining the differences in accuracy rate and error pattern between Swedish-speaking L2 Mandarin learners and Danish-speaking L2 learners. As explained previously, the listeners recruited in this study were from Nordic language backgrounds and represented a pitch-accent language (Swedish) or a non-tonal language (Danish). Comparison of their performance in the first experiment suggests that Swedish listeners outperformed Danish listeners to some extent and their perceptual difficulty varied.

Overall, both groups achieved an accuracy rate of over 80% in correctly identifying and categorizing the four Mandarin tones in isolation. The mean score of the Swedish group was nearly 90%, which was attested by the statistical analysis as performing better than the Danish group, but not significantly better. Notably, the Swedish group achieved a higher accuracy rate for all four lexical tones than the Danish group, with the differences particularly obvious for T2 and T1, as Figure 2 shows. Another relevant finding was that over half of the Swedish listeners completed this task nearly perfectly or perfectly, but that the proportion of Danish listeners who achieved this level of accuracy was much lower. This suggests that Swedish listeners, who are from a pitch-accent language background were better than Danish listeners who are from a non-tonal language background in identifying Mandarin tones.

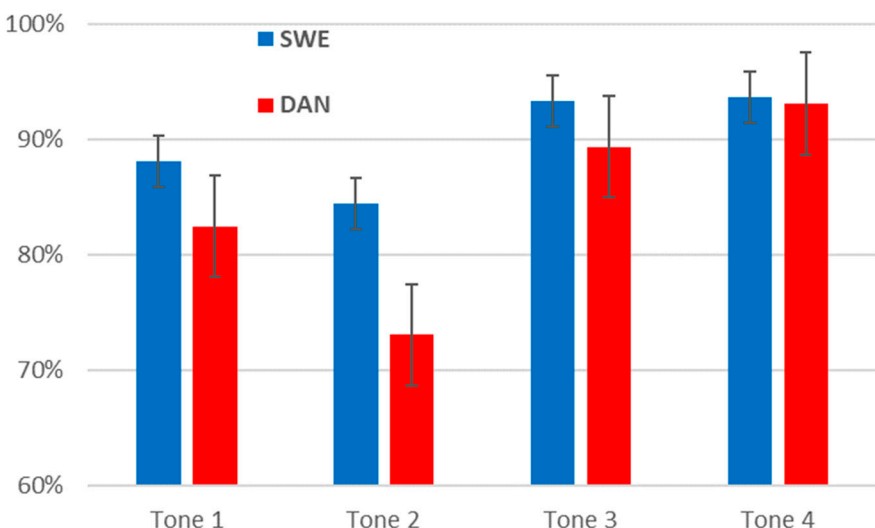

**Figure 2.** Mean accuracy scores and standard errors of the four Mandarin tones by Swedish and Danish listeners.

Comparison with previous cross-language studies, particularly So and Best (2010), reveals some consistencies, but also discrepancies with the findings of Wang (2013). So and Best (2010) reported that pitch-accent Japanese naïve listeners perceived Mandarin tones similarly to tonal Cantonese listeners, with both groups outperforming non-tonal English listeners. Conversely, Wang (2013) found that the perception of Mandarin tones by tone language (Hmong) listeners was worse than non-tonal English and pitch-accent Japanese listeners, with the latter two groups exhibiting nearly identical performance. One possible explanation for the better performance of Swedish listeners, in contrast to Wang (2013), could be attributed to the pitch accents in their native language and how the pitch accents resemble lexical tones in Mandarin. In other words, Swedish pitch accents may share more similarities with Mandarin tones than Japanese pitch accents. It is important to note, however, that differences in experimental settings between the current study and Wang (2013), as well as factors such as the listeners' Chinese learning experience and proficiency level, may also contribute to the varied outcomes.

Further analysis of the listeners' perceptual errors showed that the Swedish and Danish groups displayed different error patterns and tonal confusions. Although the perceptual difficulty ranking among the four lexical tones was the same, the effect of tone was only statistically significant for the Danish group. The major tonal confusion for Danish listeners was the T2–T3 pair, corresponding to findings in previous studies (Chen 1997; Sun 1998; So and Best 2010; Hao 2012) which showed that English-speaking listeners also confused T2 with T3 most often in both directions. As for Swedish listeners, this study revealed that the major source of confusion for them was the T1–T2 pair. This is not in full agreement with the findings from So and Best (2010). There, the pitch-accent Japanese listeners were most likely to confuse T1 with T2 (but not vice versa) and T2 with T3 in both directions. So and Best (2010) attempted to account for the different tonal confusions observed in the three language groups as language-specific error patterns within the framework of the PAM (Best 1995; So and Best 2008). Following the line of explanation in So and Best (2010), Swedish pitch accents can be used to account for the T1–T2 confusion that poses a significant challenge for Swedish listeners, a phenomenon observed exclusively in the current study. As outlined in Section 1.5, both Swedish Accent 1 and Accent 2 feature falling contours over a word, represented as HL and HLHL, respectively. Consequently, it is plausible that Swedish learners excel at recognizing tones with falling contours, particularly in the case of Mandarin, where these correspond to T3 and T4. However, for the high-level T1 and rising T2, Swedish listeners lack sufficient experience in their native language. In addition to addressing the unique T1–T2 confusion challenge, this explanation can also

provide insight into why Swedish listeners demonstrate higher accuracy scores for T3 and T4 compared with T1 and T2.

In summarizing the main implications drawn from the findings of experiment 1, two key insights emerge. Firstly, it appears that the presence of pitch accents in Swedish may assist Swedish-speaking L2 learners in accurately parsing Mandarin tones. This adds new evidence from the Nordic languages to the ongoing debate about whether the presence of tonal systems in one's native language facilitates the perception of tones in other languages. Secondly, pitch-accent Swedish learners encounter distinct challenges when processing Mandarin tones. This finding, previously unreported, constitutes new and interesting evidence, further affirming the strong influence of the L1's tonal system on the perception of non-native tones.

This first experiment helps us to understand Swedish and Danish listeners' ability to perceive tonal differences between categories. However, a second experiment was conducted to test these listeners' sensitivity to more subtle variations in pitch. Along a continuum with T1 on one end and T2 on the other end, a series of contours were created that display incremental changes in pitch. The same participants from the three language groups were then invited to categorize each stimulus as T1 or T2. This is a more challenging task when compared with the listening task in experiment 1, which aimed to address the remaining two research questions. In experiment 2, only the top performers who achieved a 90% or more overall perception score and 90% or more perception score for T1 and T2, respectively, were able to provide useful insights regarding Chinese learners' ability to detect within-category tonal variations. The choice of investigating tonal variations between T1 and T2 is based on three considerations: they differ in tone type (level tone and contour tone, respectively), thus enabling a comparison between Swedish and Danish listeners' sensitivity to flat and rising contours; they differ in duration when produced in isolation (Fu and Zeng 2000), thus allowing an examination of the effect of duration on tonal perception; and they both appeared to be challenging for Swedish and Danish groups in the first experiment.

## 3. Experiment 2

### *3.1. Methodology*

#### 3.1.1. Materials

Four T1–T2 continua with syllables /ma/ and /i/ were created as the stimuli for the second experiment. They were constructed on the basis of real speech data produced by the same female native speaker of Beijing Mandarin who produced the stimuli for experiment 1. The stimuli were created in three steps using Praat (Boersma and Weenink 2018): (a) creating exemplar contours for Tone 1 and Tone 2; (b) generating a series of F0 contours with an incrementally varied F0 between the two exemplars created in the first step; and (c) resynthesizing the syllables /mā/, /má/, /ī/ and /í/ to create four T1–T2 continua.

Ten T1 and ten T2 monosyllables (open syllables with or without a nasal onset, e.g., ma, yi, mo, wu, etc.) produced by the same female native speaker were subject to F0 analysis using the Prosody Pro script (Xu 2013). The time-normalized F0, i.e., ten F0 readings over the duration of one monosyllable, was obtained for every syllable. Subsequently, mean time-normalized F0 contours averaged across ten tokens were computed for T1 and T2; these served as the exemplars. Figure 3 displays the F0 contour of the T1 exemplar in the yellow line that starts with a small dip and stays flat till the end. It also shows the contour of the T2 exemplar in the purple line, which starts with a similar slight fall to the halfway point before steadily rising to the end. With these two contours as the ends of the continuum, an additional five contours were generated with incrementally different F0 in between these two ends (step 1 and step 7). One of these contours (step 4 stimulus) is indicated by the pink line in the middle.

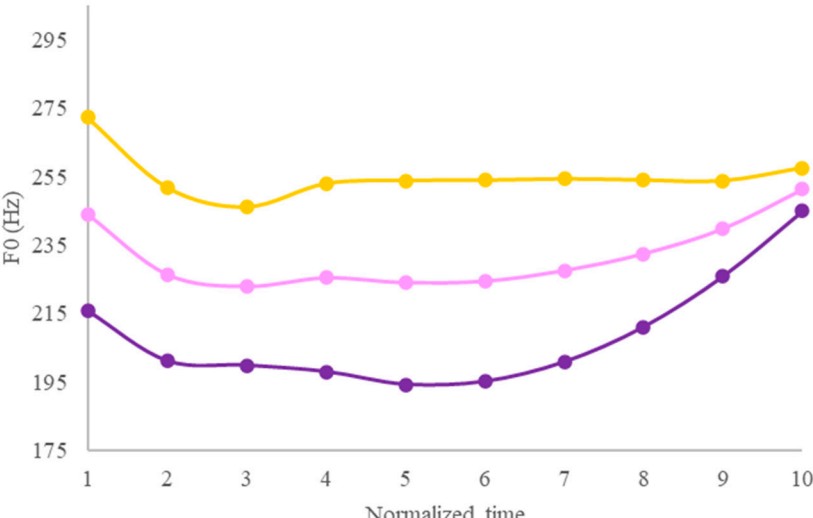

**Figure 3.** Pitch contours for the first (yellow line), fourth (pink line) and seventh (purple line) stimuli within a Tone 1–Tone 2 continuum.

Four monosyllables produced by the same female native speaker were chosen: /mā/ (T1, 347.6 ms), /má/ (T2, 420.2 ms), /ī/ (T1, 341.8 ms), and /í/ (T2, 414.8 ms). The duration difference between the T2 and T1 syllables presented in this set of stimuli is consistent with the findings from previous studies (Tseng 1990; Fu and Zeng 2000). Finally, the PSOLA (pitch synchronous overlap add) in Praat (Boersma and Weenink 2018) was used to resynthesize the stimuli, i.e., to replace the F0 contours of the selected four syllables with the contours from the seven-step continuum. Therefore, a set of 28 syllables was created for the second listening task, including seven from each of the four continua (/ma/ and /i/ with long and short durations). In each continuum, such as the long syllable /ma/, all seven syllables share identical segment-level pronunciation but differ exclusively in pitch contours. The same applies to the short /ma/ continuum, where syllables only differ from those in the long /ma/ continuum in duration. Segment-level pronunciation, voice quality, and intensity aspects remain highly similar, as evidenced by comparable intensity measures and the first two formants of the vowel for both the long and short syllables of /ma/ and /i/, respectively. As an extra control, two native speakers of Mandarin were invited to listen to the exemplars at the ends of the four continua (step 1 and step 7). Both were able to recognize the sounds as T1 or T2 and did not report any traces of unnaturalness.

### 3.1.2. Procedure

These 28 monosyllables were presented twice in separate blocks to the participants. The order of presentation was randomized differently in two blocks. Participants listened to three syllables (step 1 or step 7 of a continuum) before each block for practice purposes. Therefore, the total number of tokens presented was 62, i.e., 56 syllables (2 syllable types × 2 duration × 7 steps × 2 repetitions) and 6 fillers. Just like the first listening task, each listener completed this task independently in a quiet environment, where they listened to one syllable at a time and had 3 s to categorize it as T1 or T2 before listening to the next syllable.

All of the listeners who participated in the first listening task also took part in this task. However, for the reasons stated in Section 2.3 above, we only analyzed the performance of participants who received 90% and above accuracy rate in experiment 1, as well as for T1 and T2 separately, i.e., 15 of 27 Swedish, 10 of 32 Danish, and 16 of 16 native Chinese listeners.

*3.2. Results*

3.2.1. Identification Curves

The results of the identification task are presented in Figure 4. The comparison was carried out using the mean percentage of T1 responses; in other words, the ratio at which the listeners identified a stimulus as more like T1 (a high-level tone) than T2 (a rising tone). Comparing the curves that connect the data points in each chart, which represent one language group and syllable type, three patterns can be observed. First, the curves corresponding to the native Chinese group seem to be steeper, and thus resemble a sigmoid shape more than the curves of the other two groups. Secondly, the stimuli closer to the T2 end along the continuum, i.e., step 5 to step 7, were almost exclusively identified as T2 by the Chinese group, but this was not the case for the listeners in the other two groups. Finally, the difference between the long syllable curves (dotted lines) and the short syllable curves (solid lines) was more obvious for Swedish and Danish than Chinese listeners: the curve representing the long syllable has a steeper slope than the one corresponding to the short syllable for both syllable types. For the Mandarin group, however, the difference is less remarkable, although it presents a similar trend, especially for the syllable /i/.

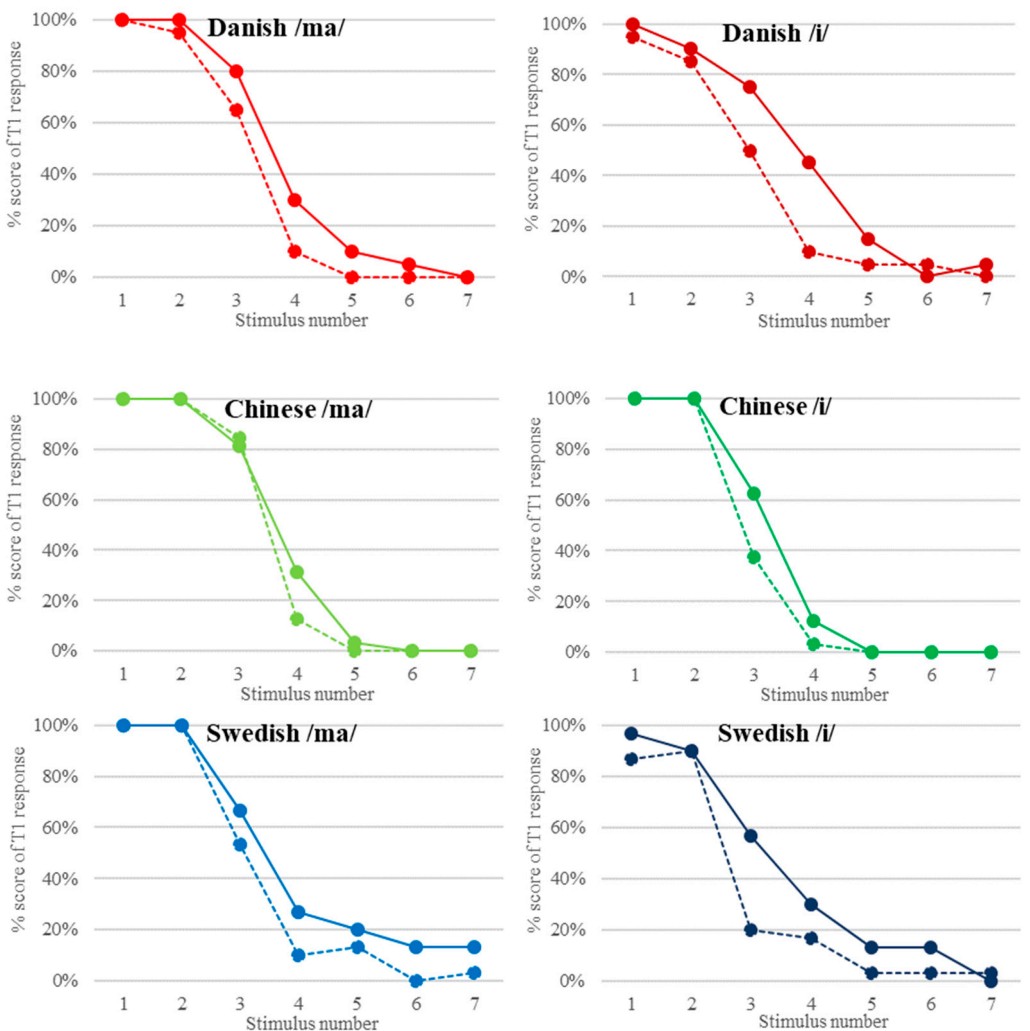

**Figure 4.** Identification curves for /ma/ (left) and /i/ (right) of Danish (top panel), Swedish (bottom panel), and Chinese (mid panel) participants. The solid lines represent the identification curves of the short syllable; the dotted lines represent identification curves of the long syllable.

3.2.2. Perceptual Boundary

This study adopted a specialized regression model to analyze how the strength of a stimulus is related to the binomial responses, namely, probit analysis (Finney 1971). It allows a more detailed analysis of the current data and has often been used in the existing literature (Hallé et al. 2004; Peng et al. 2010) to examine categorical perceptions of tones. By applying probit analysis to the results of the second listening task, we could determine the relationship between the degree of pitch contour changes and listeners' perception of tonal variations. More specifically, it computes the perceptual boundary and perceptual width for each of the identification curves in the above charts. The perceptual boundary is defined as the point at which 50% of the population categorize the stimulus as Tone 1, corresponding to the 50% crossover point on the curve.

With these conditions in mind, the perceptual boundary of the three language groups in this study appears to be similar: between three and four for the short syllables, and around three for the long syllables (see Table 2). This indicates that the perceptual boundary positions are closer to the T1 exemplar (i.e., the step 1 stimulus) for long syllables than short syllables, meaning that there were more T2 responses to the stimuli on the long syllable continua. Although syllable duration seemed to affect all three language groups' perception in the same direction, the Chinese listeners displayed only minor variations. However, for the other two groups, the effect was drastic. When we compare the two syllable types, the boundary positions of syllable /i/ are generally closer to T1 exemplar than those of syllable /ma/.

**Table 2.** Boundary positions and boundary width computed for the 3 language groups.

|  |  | /ma/-short | /ma/-long | /i/-short | /i/-long |
|---|---|---|---|---|---|
| **Danish** | Boundary position | 3.44 | 2.972 | 3.364 | 2.747 |
|  | Boundary width | 1.499 | 1.295 | 1.465 | 1.196 |
| **Chinese** | Boundary position | 3.578 | 3.406 | 3.191 | 2.874 |
|  | Boundary width | 0.963 | 0.916 | 0.858 | 0.773 |
| **Swedish** | Boundary position | 3.764 | 2.979 | 3.286 | 2.475 |
|  | Boundary width | 2.227 | 1.762 | 1.944 | 1.464 |

3.2.3. Width of the Perceptual Boundary

The perceptual boundary width for every perceptual curve, which is the distance between 25% and 75% crossover points, was also computed. This measurement often has a reverse relationship with the steepness of the curve. The width of the perceptual boundary in this case differed considerably among the three language groups in three respects. First, while the Chinese group had a narrow boundary width (less than one for all four continua), the other two groups had much broader boundary width, with some of the occasions displaying width that was nearly twice as broad as the Chinese. Second, the effect of duration on boundary width varied. While syllable length affected boundary width to a great extent for Danish listeners (between 0.2 and 0.3), it was even greater for Swedish listeners (between 0.4 and 0.5). For the Chinese group, although the perceptual curves of short syllables exhibited a broader width than long syllables, the difference was minimal (less than 0.1) when compared with the other two groups. Thirdly, syllable type also seemed to influence boundary width. The general pattern for all three groups was that syllable /i/ had a narrower width than syllable /ma/.

3.2.4. Statistical Analysis

As part of the probit analysis (Finney 1971), a regression model commonly used for analyzing binominal responses, the parallelism test, was conducted to examine whether the identification curves had a common slope. Between-group comparisons for each of the four continua were conducted using the parallelism test, and the differences among the three language groups were significant ($p < 0.01$) in all cases. When comparing the slopes

between the two learner groups only, except for the long /i/ continuum, which showed no significant difference (*p* = 0.20), significant differences were found for the other three continua at the level of *p* < 0.05.

Within-group comparisons were also made to compare the curves corresponding to the four continua. Only the Chinese group was confirmed to have a similar slope (*p* = 0.15), but both learner groups were confirmed to have significantly different slope (*p* < 0.01). The comparisons of long vs. short syllables for /ma/ and /i/, respectively, also yielded significant differences at the level of *p* < 0.01 for both learner groups.

## 4. General Discussion

The current study examined the perception of Mandarin tones by two groups of L2 Mandarin learners from distinct language backgrounds and one group of native Mandarin speakers. The learners were from a pitch-accent background—Swedish—and a non-tonal language background—Danish. The investigation was carried out in two directions: the identification of four Mandarin tones (between-category perception in experiment (1) and the T1–T2 identification (within-category perception in experiment (2)). Our findings are in good agreement with previous studies in a number of respects, but also provide fresh evidence contributing to the understanding of non-native tone perception.

The primary findings from the first experiment, aimed at addressing the first research question, revealed that, firstly, both the Swedish and Danish learner groups struggled to identify the four Mandarin tones with full accuracy, which is not unexpected. Secondly, the Swedish listeners outperformed the Danish listeners to some extent, which adds new support to the view that experience with tonal elements in one's native language may serve as a facilitative factor in processing non-native tones. However, this advantage appears to be limited and is only evident in between-category tonal differences. Swedish listeners did not display greater sensitivity to subtle within-category tonal variations, as indicated by the results of experiment 2. Finally, the Swedish and Danish groups displayed varied tonal confusion patterns, highlighting the diverse challenges encountered by listeners from different language backgrounds. More detailed discussion on these findings is presented in Section 2.3.

The second listening task (experiment 2) which, by its nature, was a different but more challenging task, yielded a number of interesting findings. For this task, the performance of some learners who achieved 90% accuracy or better in the first experiment was analyzed. The first finding from this task is that the learner groups did not perceive the within-category tonal variations in the same way as the native Chinese group. The visual and statistical comparisons of the perceptual curves corresponding to the three language groups revealed that the two learner groups had less steep curves than the Chinese group. The width of the perceptual boundary, which is often thought to correlate with the steepness of the identification curve, contributes further evidence to the general pattern: the Chinese group had a much narrower boundary width than the other two groups. This finding is consistent with the results from previous studies (Hallé et al. 2004; Peng et al. 2010; Liu et al. 2017), which reported that only native Mandarin listeners' identification curves exhibited the sigmoid shape, and thus had a steeper slope than non-native listeners' curves. The S-shape identification curve has been considered one of the classical patterns of categorical perception (Repp 1984; Hallé et al. 2004). Therefore, this finding suggests that some of the Swedish and Danish learners, despite their ability to identify the four Mandarin tones nearly perfectly, were unlikely to perceive the finer tonal variations in the same manner as native Mandarin speakers. Therefore, the answer to the second research question, investigating whether Mandarin learners who excelled in tone identification exhibit a comparable ability to perceive within-category tonal variations as native Mandarin speakers, is in the negative.

Notably, the identification of the endpoints of the continua (1 or 7 on the x-axis in Figure 4) was not always 0% or 100% for the two learners' groups, as was the case with the Chinese group. In particular, the Swedish group exhibited an identification rate of 87% for the T2 exemplar (step 7) as T1 on the short /ma/ continuum, and the T1 exemplar (step 1) as

T2 on the long /i/ continuum, a result that might be considered problematic at first glance. Close examination of the data revealed that the deviation was primarily caused by two out of fifteen Swedish listeners. Specifically, one of them identified over 85% of the stimuli along the short /a/ continuum and along the short /i/ continuum as T1, and over 85% of the stimuli along the long /i/ continuum as T2. The other Swedish listener identified all stimuli along the short /a/ continuum as T1, and 93% of the stimuli along the long /i/ continuum as T2. This outcome is closely linked to the small sampling size of the current study, which has been significantly affected by data from these two speakers. It potentially presents additional evidence suggesting that the learners who excel in between-category tonal variations may still have a substantial journey ahead to achieve native-like perception. Despite their ability to correctly identify nearly all T1 and T2 in the first listening task, these two learners may experience considerable confusion and bewilderment when exposed to syllables that exhibit very subtle differences in pitch information. Considering that the participants in this study were all beginner-level Chinese learners, this finding is in line with a prior suggestion by Zhao and Kuhl (2015): experiencing substantial exposure to the target language is crucial for enhancing learners' tonal perceptual sensitivity to a native level. Furthermore, the performance of these two learners in the second listening task is likely to be affected by the length differences of the stimuli, a point we will revisit in the upcoming discussion.

An unanticipated finding from the second listening task was how the two learner groups' performance differed. Danish listeners had a steeper identification curve than the Swedish learners, as suggested by the width of their perceptual boundary and the parallelism test. This finding is new and surprising. Earlier studies (Hallé et al. 2004; Peng et al. 2010; Liu et al. 2017) were mostly interested in comparing non-native listeners' perception of tonal variations with that from native Chinese listeners, thus there was no report of similar findings. One possible explanation for this observation is that Danish listeners are more sensitive to within-category tonal variations than Swedish listeners. While this may initially seem plausible, it is counterintuitive and is inconsistent with the key finding in the first experiment. Given their pitch-accent language background, Swedish learners should be, at the very least, as sensitive to subtle tonal variations as Danish learners, if not more. A more likely alternative explanation is that the extreme performance of the two Swedish learners, as discussed above, influenced the slope of the identification curves. To explore this, we reanalyzed the performance of the Swedish leaners after excluding data from the two extreme performers. The boundary width of the Swedish group's perceptual curves (/ma/-short: 1.416; /ma/-long: 1.266; /i/-short: 1.259; /i/-long: 1.245), negatively correlated with the steepness of the slope, became highly similar to that of the Danish group (/ma/-short: 1.499; /ma/-long: 1.295; /i/-short: 1.465; /i/-long: 1.196). Statistical tests for the between-group comparison, using the parallelism test, also suggested that the two learner groups share common slopes for all four continua at the 0.05 level ($p > 0.05$). Therefore, the explanation involving outlier participants aligns not only with the results of experiment 1, but is also supported by further data analysis, making it the more plausible and coherent explanation.

Another important finding from the second listening task was that syllable duration affected the listeners' tone perception to different extents. All three groups' responses to the short stimuli varied from the long stimuli in terms of the boundary position and boundary width. For the boundary position, continua with longer syllables were associated with smaller category boundaries. This means that the listeners tend to perceive more stimuli on these continua as T2 than the continua with shorter syllables. The same pattern was observed for all three of the language groups, but the long–short contrast was more obvious for the two learner groups. Likewise, the identification boundary width measurement displayed patterns in the same direction: continua with longer syllables had narrower boundary widths. Since the boundary width was closely associated with the steepness of the identification curve, which we mentioned earlier, it suggests that the listeners became more sensitive to the tonal variations when perceiving longer stimuli. The Swedish

and Danish groups showed a stronger association between stimuli length and boundary position and width than the Chinese group, which was further backed up with statistical tests. In the literature, only a limited number of studies have examined the duration effect on the perception of subtle tonal variations. These findings are in accordance with what has been reported previously. Blicher et al. (1990) found that syllable lengthening facilitated the perception of within-category variation between T2 and T3 by Chinese and English listeners. Chen et al. (2017) also reported the duration effect on the categorical perception of rising and falling contours by Chinese and English listeners, and claimed that duration increased the category boundary sharpness for listeners from both language groups.

Thus, to address the third research questions concerning the influence of L1 prosodic feature on the perception of within-category tonal variation, compelling evidence has been found. This evidence supports the claim that length contrast, as a prosodic feature in Swedish and Danish, significantly affects learners from these two language backgrounds to perceive T1–T2 tonal variations. As explained previously, these two Nordic languages exhibit length contrast at the segmental level, while Mandarin does not. The presence and absence of length contrast in these three languages explains the different degrees of variations associated with the long–short identification curves. The insignificance of the observed differences for the Chinese group was anticipated. The presence of length as a prosodic feature in the native languages of the two learner groups resulted in noticeable duration effects. This finding provides new evidence for the influence of linguistic experience on non-native lexical tone perception. While the influence of an L1 tonal system on the perception of Mandarin tones has been well acknowledged (So and Best 2010; Wang 2013; Tsukada 2019), the present study highlights that length contrast in L1 can also be an impact factor.

Additionally, an interesting finding emerged: the Swedish group exhibited much greater contrasts in terms of boundary position and boundary width between long and short syllables than the Danish group. The two Swedish learners' extreme performance in the second listening task, discussed earlier, provides further evidence. These two learners predominantly identify stimuli on the short syllable continuum as T1 and stimuli on the long syllable continuum as T2, considering that Mandarin T1 is consistently shorter than T2 when pronounced in isolation. While this observation suggests the intriguing possibility that the duration of stimuli might have a more pronounced influence on Swedish listeners compared with Danish listeners, caution is essential before drawing any conclusions. This difference could be attributed to the presence of length contrast in both consonants and vowels in Swedish, in contrast to Danish, which only exhibits length contrast in vowels. However, definitive insights would require further studies with a more extensive sample size.

Finally, this study has also helped us to understand the perception by L2 learners at the suprasegmental level. Even though perceiving Mandarin tones accurately has been repeatedly reported as challenging for adult learners (Kiriloff 1969; Broselow et al. 1987; Shen 1989; Hao 2012), the Swedish and Danish learners in this study, with Chinese proficiency assessed at A1 or A2 in the CEFR framework of Mandarin, demonstrated a commendable ability to identify the four Mandarin tones with high accuracy rates. In fact, many achieved near-perfect accuracy scores. This finding is not entirely surprising, as it is generally believed that sufficient contact and experience with a target language can help learners to perceive sounds in that language correctly (Best and Strange 1992; Flege 1995; Best and Tyler 2007; Hao 2012; Shen and Froud 2016; Tsukada and Han 2019). While a majority of the previous studies in this area have focused their investigations on the identification of the four Mandarin tones and the perceptual error patterns or tonal confusions that are displayed, little attention has been paid to language learners' ability to sense subtle differences within a tonal category, except Shen and Froud (2016). Advanced learners of Mandarin Chinese, as reported in this study (Shen and Froud 2016), with a minimum of three terms' study, demonstrated the potential to perceive Mandarin tones categorically, similar to native Mandarin speakers. The current results show that, even

though beginner-level Mandarin learners could perceive the phonetic properties of the four lexical tones nearly perfectly, they were unable to perceive the within-category variations between T1 and T2 in the same manner as native Mandarin Chinese speakers. Within the theoretical frameworks of second-language speech perception (Flege 1995, 2007; Best and Tyler 2007), this phenomenon could be attributed to the learners not having established new phonetic categories for these novel lexical tones. In the realm of second-language learning, it is crucial for learners to continue to receive abundant natural language input and targeted training. This ongoing exposure and specialized instruction are essential for acquiring non-native tones in a manner that approximates native-like proficiency. The findings of this study could also serve as a basis for future research efforts aimed at improving learners' perceptual sensitivities to non-native tones, with the goal of achieving a proficiency level that closely resembles native perception.

## 5. Limitations and Future Directions

The findings presented in this study offer valuable insights into the perception of Mandarin tonal variations by L2 learners from Nordic language backgrounds. However, it is crucial to acknowledge certain limitations that may impact the generalizability of the results. The relatively small sample size, particularly in experiment 2, poses a limitation. Additionally, the absence of a discrimination task to supplement the categorical perception results is another limitation that warrants consideration.

This study highlights potential avenues for future research. Firstly, the identified limitations emphasize the need for a more extensive study with a larger sample size, as well as the inclusion of additional listening tasks to investigate perception to within-category tonal changes by learners from Nordic backgrounds. Future research could also involve recruiting L2 Chinese learners at different proficiency levels to examine their sensitivity to subtle tonal variations, contributing to our understanding of whether and how L2 learners can achieve native-like proficiency. Lastly, the outcomes of the first experiment provide a foundation for a more detailed examination, addressing the T1–T2 confusion that poses major challenge Swedish learners and exploring how Swedish pitch accents influence learners' perception of the four Mandarin tones.

## 6. Conclusions

In this study, the aim was to examine the perception of Mandarin tonal variations by L2 learners from a pitch-accent language (Swedish) and a non-tonal language (Danish) background, and, in addition, to assess the phonetic and phonological effects of learners' native language on tone perception. The main conclusions drawn from the results are as follows: (1) Swedish-speaking learners outperformed their Danish-speaking counterparts in identifying the four Mandarin lexical tones. In addition, the T1–T2 pair presents a great challenge for Swedish learners, while Danish learners faced major confusion with the T2–T3 pair; (2) Neither learner group is able to perceive the subtle within-category tonal variations in the same way as native Mandarin speakers; (3) The prosodic feature of an L1, the length contrast in particular, has a considerable influence on the perception of within-category tonal variations in Mandarin Chinese. To the best of our knowledge, this is the largest study published to date on tone perception by learners from Nordic language backgrounds. These findings add to the growing body of research that confirms the native phonological influences on the perception of non-native tones. This article also reports an initial attempt towards a more profound understanding of the within-category perception of tonal variations by L2 learners who excel at between-category perception.

**Funding:** This research received no external funding.

**Institutional Review Board Statement:** Not applicable.

**Informed Consent Statement:** Informed consent was obtained from all participants involved in the study.

**Data Availability Statement:** The data presented in this study are available on request from the author.

**Acknowledgments:** I extend my sincere gratitude to all the participants for their valuable contributions to this study. I am particularly grateful to Chun Zhang for her support in recruiting participants, as well as her consultation and comments on this study. Additionally, I would like to express my appreciation for the insightful feedback provided by the anonymous reviewers and editors, which significantly enhanced the quality of this paper.

**Conflicts of Interest:** The author declares no conflicts of interest.

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
