# Peer review of "Cross-Language Perception of Lexical Tones by Nordic Learners of Mandarin Chinese"

_languages, doi:10.3390/languages9020065_

Round 1
Reviewer 1 Report
Comments and Suggestions for Authors
Dear authors,
Thank you for the opportunity to review this article. This article describes a study of the differences in tonal detection/discernment among Swedish and Danish speakers. It is novel in the aspect that Swedish uses pitch whereas Danish does not. The authors postulate that the participants' L1 will affect the outcome of the study as L1 is a known factor in the perception of a novel L2.
Overall, the paper is extremely well-written and polished. The literature review is coherent and relevant. The justifications for the study and the procedures are robust. The study design is also satisfactory. I particularly like that care was taken to assure stimuli were composed of sounds present in both languages and that existed in all four tones in Mandarin (pg. 6).
However, one major concern I had with the article is that of data presentation and analysis, particularly with respect to Experiment One. Since the RQ wants to compare the performance between Swedish and Danish speakers, why was the ANOVA performed within-group only? Visual analysis alone is insufficient to detect if there were statistical differences between the groups and therefore, subsequent discussion based on that conclusion are (at present) unfounded. The author notes, "...was attested by the statistical analysis as performing better than the Danish group" (pg. 7), but no such analysis was reported in my understanding.
In addition, it would be typical to display results in tabular form, rather than relying solely on exposition in the text.
General comments:
・"The listening stimuli included 40 tokens of ten syllables..." is misworded. The 40 tokens were of ONE syllable each, were they not?
・Greek letters (X, p, etc.) should be italicized. Statistics themselves should be rounded off to 2 decimal places.
・"...in separate blocks to the INFORMANTS" (pg. 10) is this a typo? Should this be PARTICIPANTS?
・"p=0.000" (p. 12) is incorrect. It should be p<.01.
・The article is missing sections of Limitations & Future Directions.
Author Response
Response to Reviewer 1 Comments for Manuscript ID [languages-2757394]
I would like to express my sincere gratitude to Reviewer 1 for the thorough examination and valuable feedback on my manuscript. I have carefully considered each comment, and I am pleased to submit the revised manuscript along with this response letter.
- However, one major concern I had with the article is that of data presentation and analysis, particularly with respect to Experiment One.
Response: The results of Experiment 1 have been subjected to a reanalysis of the statistical analysis, both between-group differences and within-group differences have been tested using ANOVA. The outcomes of these statistical tests are presented in section 2.2 (line 297 to 310).
- It would be typical to display results in tabular form, rather than relying solely on exposition in the text.
Response: Table 1 (page 7) has been added to display the results of error confusion in Experiment 1.
- The listening stimuli included 40 tokens of ten syllables..." is misworded. The 40 tokens were of ONE syllable each, were they not?
Response: You are correct! Each token in the listening stimuli is one syllable, the expression has been rephrased (line 266-269).
- Greek letters (X, p, etc.) should be italicized. Statistics themselves should be rounded off to 2 decimal places.
Response: Modifications have been incorporated throughout this paper based on your suggestion. Thank you for bringing it to my attention!
- "...in separate blocks to the INFORMANTS" (pg. 10) is this a typo? Should this be PARTICIPANTS?
Response: ‘Participants’ is indeed a more suitable term for this study, which reports two perception experiments. Changes have been made throughout this paper as suggested.
- p=0.000" (p. 12) is incorrect. It should be p<.01.
Response: Changes have been made accordingly.
- The article is missing sections of Limitations & Future Directions.
Response: In the revised manuscript, Section 5, covering limitations and future directions, has been included (line 704 to line 720).
I hope that the revisions made to the manuscript have enhanced the clarity of result presentation and the overall quality of paper. I am confident that these changes contribute to the rigor and clarity of the manuscript. Once again, thank you for your constructive comments and the opportunity to improve the manuscript.
Reviewer 2 Report
Comments and Suggestions for Authors
The present study investigates the perception of Mandarin tone by Swedish and Danish learners of Mandarin as well as native Mandarin speakers. The choice of Swedish and Danish is interesting because the former, being a pitch-accent language, uses F0 contrastively while the latter does not. Two perception experiments were conducted. In Exp. 1, listeners are presented with natural Mandarin syllables and indicate which tone they heard. In Exp. 2, listeners are presented with stimuli from multiple T1-T2 continua and again indicate which tone they heard. Exp. 1 provides a basic idea of tone identification accuracy across the three listener groups. Exp. 2 seeks to provide a more fine-grained analysis that allows for a comparison of within-category sensitivity. For this experiment, only those listeners with an accuracy of 90% or above in Exp. 1 were included.
Results from Exp.1 showed that Swedish listeners outperformed Danish listeners, suggesting that the native pitch-accent system helped their L2 tone perception. Exp. 2 shows that despite their very accurate tone identification in Exp. 1, non-native listeners differ from native Mandarin listeners. Specifically, non-native identification functions are less steep and non-native listeners show an effect of tone duration which native listeners do not.
The idea of comparing listeners from a pitch-accent language to those from a non-pitch accent language is interesting but I am not sure that the present study makes a sufficient additional contribution to our understanding of L2 tone perception. It is not clear how the present results relate to those from studies mentioned in the Introduction that compared performance between listeners from tone (or pitch-accent) languages and listeners from non-tonal languages. We now seem to have conflicting findings, with some studies showing no effect of the L1 tonal status, and other studies (including the present one) finding such an effect. Without at least some tentative explanation of these discrepancies (difference in methodology, L1?), the current results are difficult to interpret.
Below, I list specific comments in order of the manuscript.
l. 28 Insert “L2” in between “the” and “perception”
Section 1.3 The authors seem to contrast perception that is categorical from perception that is either “psychophysical” (l. 132) or “psycho-acoustic” (l. 141). Is there a difference between the latter two terms? If so, it should be made clear. If not, it may be better to only use one of these terms throughout.
l. 157-168 I think RQ 1 could be broader (do speakers of an L1 with a pitch-accent system differ from those of an L1 without tone or pitch-accent)?
The inclusion of phonemic vowel length seems ad hoc. Ultimately, is the idea that this phonemic distinction (which Mandarin lacks) would make Nordic listeners more sensitive to duration?
l. 234 Danish consonants do not “seem” to contrast in length. Unclear – either they do, or they don’t.
l. 237 The use of “informants” seems odd. Replace with “participants” throughout?
l. 242 Did none of the Danish participants have any knowledge of Swedish?
l. 247 Please provide more information about these classifications. The CEFR website lists A1 and A2 as “basic user”. This would be good to mention.
l. 280 Not sure why an ANOVA wasn’t conducted instead of a t-test. Were the Swedes significantly different from the Mandarin participants?
l. 296 Insert “common” between “most” and “tonal”.
l. 299 Please provide the entire confusion matrix. Also, many studies report T2 and T3 as most confusable. The present results find T1 and T2 most confusable. Why this difference? Is it perhaps due to the particular productions used in this study? Figure 3 also suggests that this speaker’s T1 and T2 are perhaps not as “canonical” as those used in other studies?
l. 330 Is the better performance by the Swedes due to their overall greater sensitivity to F0 or to the fact that some of the Mandarin tones can be mapped onto the Swedish pitch accents?
Section 3.3.1 I am confused by the duration manipulation of the stimuli. How are there two durations for each continuum? Taking /ma/ as an example, is it the case that one continuum took T1 as the basis and all continuum members were 347.6 ms? And the other took T2 as the basis and all members were 420.2 ms?
If that’s the case, and short stimuli are based on T1 productions and long stimuli on T2 productions, how confident are the authors that duration is the main difference between the stimuli, as compared to, say, voice quality or intensity?
l. 490 Again, not sure why ANOVA wasn’t used to compare all three language groups?
l. 536 I object to labeling the L2 participants in Exp. 2 as “proficient”. All participants were previously described as basic users (A1, A2). Now, the criterion seems to have shifted and those participants with >90% accuracy in Exp. 1 are considered proficient??
l. 568 What is the explanation for the “counterintuitive” finding that the Danes show greater within-category sensitivity than the Swedes? The authors need to at least provide a tentative explanation. It would seem contradictory to their explanation for Exp. 1, that the presence of a pitch-accent distinction makes the Swedes more sensitive to differences in F0. Without an explanation of this Danish advantage, we are left with a pattern of results that is difficult to interpret.
If the two Swedish outliers caused this difference, it may be worthwhile to show what happens when these two participants are excluded from the analysis.
Comments on the Quality of English Language
N/A
Author Response
Response to Reviewer 2 Comments for Manuscript ID [languages-2757394]
I would like to express my sincere gratitude to Reviewer 2 for the thorough examination and extremely valuable feedback on my manuscript. I have carefully considered each comment, and I am pleased to submit the revised manuscript along with this response letter.
- It is not clear how the present results relate to those from studies mentioned in the Introduction that compared performance between listeners from tone (or pitch-accent) languages and listeners from non-tonal languages. We now seem to have conflicting findings, with some studies showing no effect of the L1 tonal status, and other studies (including the present one) finding such an effect. Without at least some tentative explanation of these discrepancies (difference in methodology, L1?), the current results are difficult to interpret.
Response: Thank you for highlighting the crucial weakness regarding clarity in presenting how the current results relate to previous studies. In response to your suggestion, I have made significant improvements in this regard. The discussion part (section 2.3) has been rewritten to provide a clearer comparison between the current findings and those of previous relevant studies (line 345 to 351). Subsequently, an explanation has been offered to account for the discrepancies observed between the results of the current study and another study (Wang, 2013) (line 352 to 358).
- Insert “L2” in between “the” and “perception”
Response: Insertion has been made (line 28).
- Section 1.3 The authors seem to contrast perception that is categorical from perception that is either “psychophysical” (l. 132) or “psycho-acoustic” (l. 141). Is there a difference between the latter two terms? If so, it should be made clear. If not, it may be better to only use one of these terms throughout.
Response: The term ‘psycho-acoustic’ is a subset of ‘psychophysical’, although in the context of language perception studies, they essentially refer to the same concept. Following a thorough review of relevant literature, it is noted that 'psychophysical' is slightly more prevalent; therefore, this term has been consistently chosen for use throughout.
- I think RQ 1 could be broader (do speakers of an L1 with a pitch-accent system differ from those of an L1 without tone or pitch-accent)?
Response: Research question 1 has been rephrased, which is now more consistent with the other two research questions (line 161 to 164). Thank you for bringing this to my attention!
- The inclusion of phonemic vowel length seems ad hoc. Ultimately, is the idea that this phonemic distinction (which Mandarin lacks) would make Nordic listeners more sensitive to duration?
Response: Yes, it is very important to ensure that every research question is well-founded and aligned with the overarching aim of the paper. Your insight is really appreciated, and I agree with your interpretation regarding this question. In response, I have incorporated additional lines to explicitly articulate the relevance of this research question. (line 168 to 175)
- Danish consonants do not “seem” to contrast in length. Unclear – either they do, or they don’t.
Response: ‘Seem’ is removed from the text. (line 242).
- The use of “informants” seems odd. Replace with “participants” throughout?
Response: ‘Participants’ is indeed a more suitable term for this study, which reports two perception experiments. Changes have been made throughout this paper as suggested.
- Did none of the Danish participants have any knowledge of Swedish?
Response: Information has been provided regarding Danish participants’ knowledge of Swedish (line 252 to line 255).
- Please provide more information about these classifications. The CEFR website lists A1 and A2 as “basic user”. This would be good to mention.
Response: More information about CEFR level A1 and A2 has been added. (line 260 to line 262).
- Not sure why an ANOVA wasn’t conducted instead of a t-test. Were the Swedes significantly different from the Mandarin participants?
Response: The results of Experiment 1 have been subjected to a reanalysis of the statistical analysis, both between-group differences and within-group differences have been tested using ANOVA. The Mandarin group performed significantly better than the Swedish and Danish groups. The outcomes of these statistical tests are presented in line 297 to 310.
- Insert “common” between “most” and “tonal”.
Response: Insertion has been made (line 315).
- Please provide the entire confusion matrix. Also, many studies report T2 and T3 as most confusable. The present results find T1 and T2 most confusable. Why this difference? Is it perhaps due to the particular productions used in this study? Figure 3 also suggests that this speaker’s T1 and T2 are perhaps not as “canonical” as those used in other studies?
Response: Table 1 (page 7) has been added to display the results of error confusion in Experiment 1.
The speech material employed in the perception experiments was produced by a female speaker, a native of Beijing Mandarin. The naturalness and native-like quality of the speech stimuli used in both experiments were confirmed through evaluation by two native speakers. An explanation for the unique T1-T2 confusion observed among Swedish listeners has been incorporated into the revised manuscript. (line 371-381)
- Is the better performance by the Swedes due to their overall greater sensitivity to F0 or to the fact that some of the Mandarin tones can be mapped onto the Swedish pitch accents?
Response: This is indeed a question to the point! The results from experiment 1 suggest that Swedes are more sensitive to between-category tonal variations, enabling them to better identify tones from four distinct tonal categories in Chinese. However, this does not extend to within-category differences, as revealed in experiment 2. In response to Research Question 1, I have rephrased the summary in section 4, and provided a brief discussion on this point (line 546-557). The mapping of Mandarin tones onto Swedish pitch accents has been introduced to account for the distinctive T1-T2 confusion observed among Swedish listeners. These discussions have been integrated into section 2.3. (line 371-381)
- Section 3.3.1 I am confused by the duration manipulation of the stimuli. How are there two durations for each continuum? Taking /ma/ as an example, is it the case that one continuum took T1 as the basis and all continuum members were 347.6 ms? And the other took T2 as the basis and all members were 420.2 ms?
If that’s the case, and short stimuli are based on T1 productions and long stimuli on T2 productions, how confident are the authors that duration is the main difference between the stimuli, as compared to, say, voice quality or intensity?
Response: In Experiment 2, it is correct that all tokens within each continuum have the same segment-level pronunciation. Therefore, yes, the interpretation is right that for short /ma/ continuum, all seven tokens are from the same syllable and are, therefore, 347.6 ms long, differing only in pitch contours. The syllables for short /ma/, long /ma/, short /i/ and long /i/ were produced by the same speaker in a single recording session, and meticulous attention was devoted to the stimulus selection process, I am confident that the chosen syllables exhibit a high degree of similarity in segmental-level pronunciation, voice quality and intensity. The description of the stimuli for Experiment 2 has been partially reworded to improve the clarity. (line 440 to line 446).
- Not sure why an ANOVA wasn’t conducted instead of a t-test. Were the Swedes significantly different from the Mandarin participants?
Response: Given the specific nature of the results from Experiment 2, involving binomial responses, probit analysis is more suitable than ANOVA tests. Within the specialized regression model of probit analysis, the parallelism test was employed to assess whether the identification curves of the three groups share the same slope. The establishment of parallelism (p > 0.05) indicates a shared sensitivity among these three groups; conversely, a lack of parallelism suggests differing sensitivities. Importantly, the t-test results, originally presented in the manuscript, were found to be irrelevant to the comparison of slopes. As a result, the statistical analysis in section 3.2.4 has been redrafted. (line 523 to line 536)
- I object to labeling the L2 participants in Exp. 2 as “proficient”. All participants were previously described as basic users (A1, A2). Now, the criterion seems to have shifted and those participants with >90% accuracy in Exp. 1 are considered proficient??
Response: The term “proficient” was originally intended to denote participants who achieved near perfect or perfect scores in Experiment 1. Recognizing the potential for confusion associated with this term, instances where it was used have been rephrased.
- What is the explanation for the “counterintuitive” finding that the Danes show greater within-category sensitivity than the Swedes? The authors need to at least provide a tentative explanation. It would seem contradictory to their explanation for Exp. 1, that the presence of a pitch-accent distinction makes the Swedes more sensitive to differences in F0. Without an explanation of this Danish advantage, we are left with a pattern of results that is difficult to interpret.
If the two Swedish outliers caused this difference, it may be worthwhile to show what happens when these two participants are excluded from the analysis.
Response: I greatly appreciate the suggestion to conduct another analysis of the Swedish data by excluding the outlier data. A reanalysis has been performed, and the results are presented in section 4. More importantly, the reanalysis offers substantial evidence supporting one explanation for the observed pattern, as opposed to the rather counterintuitive alternative (line 608 to line 624). I sincerely value this very insightful and to-the-point comment!
I hope that the revisions made to the manuscript have enhanced the overall quality of paper, particularly in terms of data analysis, coherence, and compelling nature of the discussion sections. Once again, thank you for your constructive comments, great advice, and valuable input for improvement!
Round 2
Reviewer 1 Report
Comments and Suggestions for Authors
Dear Author,
     I congratulate you on your perseverance and effort made in your revisions. The clarity of your presentation and the quality of your discussions have thus greatly improved. I have no objections to your work, except for one statement on pg. 7, Line 300-1. "However, the difference between the performance of Swedish and Danish learners was marginally different (p = 0.08)". 
This statement is factually incorrect. In the case of a p that is greater than 0.05, the means of the two groups are statistically equal. In other words, it can be concluded that the results appeared by chance. Therefore it is inaccurate to present the two groups as different. (You correctly specified that two means w/ p =0.20 were equal/not significantly different later on in the text; p =0.08 should be treated similarly.)
Author Response
Response to Reviewer 1 comments for resubmission [languages-2757394]:
Thanks to Reviewer 1 for reviewing my resubmission and providing additional comments on lines 300-1. I appreciate your input, and the necessary change has been made accordingly (line 301-303).
Reviewer 2 Report
Comments and Suggestions for Authors
I am happy with these revisions. I see one remaining issue:
I had previously asked how confident the author is that duration is the main difference between the stimuli, as compared to, say, voice quality or intensity given that the short stimuli are based on T1 productions and long stimuli on T2 productions.
The author has responded that they are confident that "segment-level pronunciation, voice quality, and intensity aspects remain highly similar, as they were produced by the same speaker." (l. 445-447). But there may be intrinsic differences in these aspects between T1 and T2 productions. Having the same speaker produce all stimuli does not control for such natural intrinsic differences. I would therefore urge the author to conduct an acoustic analysis to prove (rather than speculate) that there are no such differences.
Comments on the Quality of English Language
The English is very good. There is the occasional typo or absence of a determiner.
Author Response
Response to Reviewer 2 comments for resubmission [languages-2757394]
Thanks to Reviewer 2, for your thorough review of my resubmission and for providing additional comments, particularly on the intrinsic differences of the stimuli used in experiment 2. I appreciate your suggestion to perform an acoustic analysis of the long-short syllable pairs, and I have conducted this analysis. The mean intensity of the syllables, as well as the F1 and F2 values at the midpoint of each syllable’s vowel, were measured and compared for each pair, offering more convincing evidence for the similarity of the stimuli. Thank you once again for yet another good suggestion! Moreover, I have carefully reviewed the paper again, and identified a typo, along with a couple of places with missing articles. I have made the necessary changes accordingly.